# Mueller Matrix Polarimetry on Cyanine Dye *J*-Aggregates

**DOI:** 10.3390/molecules28041523

**Published:** 2023-02-04

**Authors:** Samuel R. Clowes, Dora M. Răsădean, Tiberiu-M. Gianga, Tamás Jávorfi, Rohanah Hussain, Giuliano Siligardi, G. Dan Pantoş

**Affiliations:** 1Department of Chemistry, University of Bath, Claverton Down, Bath BA2 7AY, UK; 2B23 Beamline, Diamond Light Source, Didcot OX11 0DE, UK

**Keywords:** *J*-aggregates, Mueller matrix polarimetry, cyanine dyes, circular dichroism

## Abstract

Cyanine dyes are known to form *H*- and *J*-aggregates in aqueous solutions. Here we show that the cyanine dye, S0271, assembles in water into vortex induced chiral *J*-aggregates. The chirality of the *J*-aggregates depends on the directionality of the vortex. This study utilised both conventional benchtop CD spectropolarimeters and Mueller matrix polarimetry. It was found that *J*-aggregates have real chirality alongside linear dichroism and linear and circular birefringence. We identify the factors that are key to the formation of metastable chiral *J*-aggregates and propose a mechanism for their assembly.

## 1. Introduction

Cyanine-based dyes consist of two quaternised nitrogen-containing heterocycles linked by a polymethine bridge [1]. It is well known that these molecules have the ability to organise into supramolecular assemblies; depending on the orientation of the dipole moments, *H*- or *J*-aggregates are formed. *J*-aggregates are characterised by bathochromically shifted narrow absorption bands with respect to monomer absorption, whereas hypsochromically shifted bands correspond to *H*-aggregates [2,3,4,5,6,7,8]. Most of the dyes are intrinsically achiral, and their molecular energy levels can be perturbed by the experimental conditions (e.g., concentration, additives, solvent), thus changing their photophysical properties. The formation of optically active *J*-aggregates in the presence of chiral templates has been explored in the literature [3,9,10,11]. It has also been shown that, in solution, flow dynamics spontaneously generate enantiomerically enriched *J*-aggregates [12,13,14]. This macroscopic symmetry breaking during the self-assembly process imposes the formation of one *J*-aggregate enantiomer over the other [15]. Although this behaviour enhances the interest shown in cyanine dyes, its origins have been questioned and associated with other processes. The first evidence of enantiomerically enriched *J*-aggregates of cyanine dyes formed in the absence of a chiral template dates back to the 1970s [12]. Chiral *J*-aggregates were obtained by “regular stirring of the solution” and assessed by CD spectroscopy. However, the origins of this spontaneous chiroptical phenomenon have been contradicted by others. The extrinsic chirality measured was ascribed to artefacts such as birefringence and linear dichroism (LD) [15,16]. On the other hand, more recent studies using Mueller matrix polarimetry (MMP) on pseudo isocyanine chloride but also on porphyrin-based *J*-aggregates have shown that the chiral response is not an artefact, but rather a natural optical behaviour [17]. Additionally, laminar flows in solution can lead to hydrodynamic forces that influence the species in solution. Therefore, self-assembly behaviour under laminar flow will be different to that observed under Brownian dynamics [17,18,19]. Cyanine-dye-based *J*-aggregates have been explored for their use in plasmonics as excitonic scaffold alternatives to noble metals due to their tuneable chiroptical properties [20,21,22]. This work aims to address some of these concerns.

We report the properties of chiral *J*-aggregates formed by organisation of an intrinsically achiral cyanine dye monomer, 5-phenyl-2-[2-[[5-phenyl-3-(4-sulfobutyl)-3*H*-benzoxazol-2-ylidene]-methyl]-but-1-enyl]-3-(4-sulfobutyl)-benzoxazolium hydroxide (commonly referred to as S0271) under the influence of flow dynamics (Figure 1). The origin of this aggregate formation via supramolecular polymerisation is also discussed. The chiral properties of these *J*-aggregates are first discussed in terms of the observed circular and linear dichroism (CD and LD) signal using benchtop spectropolarimeters and the Diamond B23 MMP beamline to extract the CD chiroptical properties from LD, LB and CB (linear and circular birefringence) contributions in solution and in the solid state. This analysis will add unprecedented information about the self-assembly of cyanine dyes. Here, we aim not only to bridge the gap between contradictory views reported in the literature [12,15,16,23], but also to provide valuable information on how to control the formation of these mechanically induced chiral supramolecular polymers. To the best of our knowledge, the literature lacks knowledge about the polymerisation mechanism of cyanine dyes—key for a complete understanding of chiroptical processes occurring during *J*-aggregate formation. This work paves the way towards the transfer of chiral information of *J*-aggregates from solution to solid state, and therefore their potential use as an alternative to plasmonics [20,22].

## 2. Results and Discussion

### 2.1. Solutions and Preparation

The literature lacks a comprehensive list of elements that need to be considered when attempting to form ordered and reproducible supramolecular assemblies of cyanine dyes. We have considered the solvent, temperature, concentration, volume, stirrer dimensions and shape as well as stirring speed and direction as factors that can impact the formation of *J*-aggregates. For more information on the particulars of the experiments, please refer to the Materials and Methods in Section 3.1.

### 2.2. J-aggregate Assembly Mechanism

There are two main mechanisms for the formation of supramolecular polymers reported in the literature: isodesmic and cooperative [24], each having unique features that allow us to distinguish between them. Variable concentration or temperature studies allow us to discriminate between these types of supramolecular polymerisation mechanism. We have opted for the latter as the formation of *H*-aggregates [2,3,4,6,7,8] during concentration studies interferes with the data analysis pertinent to the formation of *J*-aggregates [2,3,4,6]. There is a precedent in the literature that reports the temperature-dependence behaviour of *J*-aggregates of cyanine dyes [14,25,26,27]. However, these articles do not discuss the mechanisms behind the polymerisation processes, which has prompted us to look into this. Aqueous solutions of different concentrations (20 and 50 µM) at two pH values (4 and 6.7) for S0271 were subjected to variable temperature (VT) studies using absorption spectroscopy. *J*-aggregate formation was also induced by pre-stirring the solutions over four hours at 400 rpm (detailed discussion about the stirring influence is provided in the second part of this manuscript).

Degradation of the assemblies was monitored as a function of temperature at the maxima corresponding to the *J*-aggregate band, which is 562 nm for S0271. The melting curve of S0271-based supramolecular assemblies has a sigmoidal profile, indicating a direct transition from aggregated to non-aggregated state. Therefore, the formation of *J*-aggregates of S0271 under these particular conditions is driven by an isodesmic polymerisation mechanism, in which the association constant for the addition of molecule *n* + 1 to a stack of *n* molecules is constant (Figure 2). The depletion of both CD and absorption responses as the temperature increases shows that the *J*-aggregate disassembles upon temperature increase.

### 2.3. Mechanically Induced Chirality of J-aggregates

#### Studies on Benchtop Spectropolarimeters

We performed both CD and LD studies on pre-stirred and in situ stirring solutions of cyanine dyes, which have suggested that vortex stirring generates a hydrodynamic torque at the supramolecular level. The direction (i.e., clockwise—CW or counterclockwise—CCW) imposes the induced CD sign: positive for a CCW stirring vortex and negative for a CW one, with a subsequent smaller peak of opposite sign. This is shown in Figure 3 for S0271, where the vortex preferentially orientates *J*-aggregate formation depending on stirring direction (same dye concentration in each case). The sample that has not been stirred shows no CD. The direction of the vortex imposes the chirality directionality, with the symmetry spontaneously breaking during the self-assembly of the monomers into *J*-aggregates. This hypothesis, discussed in the literature for other *J*-aggregate-forming molecules such as porphyrins, is used to explain the induction of mechanochirality [17,28]. The flow shear within a solution causes the particles to rotate, generating a shear gradient. The particles start to impose a fold and twist in the *J*-aggregate, eventually forming an enantiomeric excess of chiral folds in a helical arrangement.

A concentration-dependence study was undertaken with the aim to discover whether the intensity of the CD response is dependent on concentration. As detailed in Table 1 and Appendix A, for concentrations up to 50 μM, the dissymmetry factor, g_abs_, is constant within the experimental error. Over this concentration, g_abs_ decreases, likely due to the formation of *H*-aggregates, supported by an increase in the absorbance band at 475 nm.

In addition to concentration-dependence studies, kinetics experiments were performed on a benchtop CD spectrometer using its built-in stirring functionality. These experiments implied continuous stirring of a freshly made solution (20 μM, no pre-stirring) in the CD instrument for six and a half hours (500 rpm, CW rotation). A spectrum was recorded every ten minutes, and the data were plotted as a function of CD (and absorption) change over time (Figure 4). The CD profile this time displayed a moderate bisignate band that also remained constant throughout, without changing after further stirring or diminishing without stirring. The bisignate CD band is more apparent in the kinetics experiment, as a different shaped cuvette and stirrer bar size were used when compared to the other measurements. These experiments show that the generation of *J*-aggregates is almost instantaneous upon the handedness of stirring as the *J*-aggregate band is observed during the acquisition of the first spectrum. The solutions retained the chiral *J*-aggregates even after five days, regardless of whether they were left stirring or not (Appendix A). This has suggested that the dye molecules under the flow shear rates have a plastic-like behaviour; therefore, the chiral *J*-aggregate arrangement is long lasting. These thermodynamically favoured self-assembled structures have “memorised” an induced metastable chirality.

The shape and method of assembly of the S0271 *J*-aggregates raise questions regarding the nature of the observed CD signal using benchtop spectropolarimeters, as contributions from LD and LB to the observed spectra are difficult to assess. For solutions under constant laminar flow, LD contributions can be identified, as the magnitude of the signal is dependent on the alignment of the *J*-aggregates with the direction and speed of the laminar flow. We performed these experiments using a solution of S0271 at 20 µM that was pumped at four different constant rates through a flow cell using a peristaltic pump. The CD and LD spectra were acquired and compared with the corresponding spectra of identical solutions but under no laminar flow. Our data indicated that under laminar flow conditions there is a significant LD contribution to the observed signal in the benchtop spectropolarimeters. As the flow rate is increased for both stirred (CW, in situ) and non-stirred solutions, the LD signal has increased accordingly (Figure 5).

### 2.4. Mueller Matrix Polarimetry Studies

As described above, the observed signal in benchtop CD spectrometers is not a “pure” differential absorbance of left- vs. right-handed circularly polarised light because the solutions of the rod-like *J*-aggregates do not lead to an isotropic optically active medium [15,16,17]. To comprehensively assess the chiroptical properties of these solutions, we have taken advantage of the Mueller matrix polarimeter (MMP) facility available at the B23 beamline at the Diamond Light Source, UK. The MMP contains four photoelastic modulators (PEMs) which allow the simultaneous measurement of all the Mueller elements [29,30]. Thus, the MMP can break down what would be perceived as a CD signal in a commercial benchtop spectropolarimeter into components of LD, CB, LB, as well as true CD, along with the sample’s absorption.

The MMP spectra of S0271 solutions were recorded for samples at 15, 25 and 35 µM concentrations with both CW and CCW stirring senses. The B23 MMP setup allows scanning of the same sample using either MMP or CD mode without the need to remove the sample from the sample holder. This ensured that the same sample position was maintained throughout the experiments. The CD spectrum for all samples was recorded in this setup prior to the acquisition of MMP data, maintaining the same parameters for both runs. The MMP data represent an average of four runs.

A representative output from the MMP is presented in Figure 6. There are seven components which are measured by the MMP (CD: circular dichroism, CB: circular birefringence, LD: horizontal linear dichroism, LB: horizontal linear birefringence, LD’: 45° linear dichroism, LB’: 45° linear birefringence and A: absorbance) compared to the two (CD and A) measured by a commercial benchtop spectropolarimeter. The processing software developed at B23 for the analysis of MMP data outputs the dissymmetry g-factor (g_abs_) and transmission in addition to the seven elements described above [23,30]. In the transmission graph, the transmissions of all four runs are plotted in order to test whether deattenuation or depolarisation effects are present in the data set (effects which would make the data more difficult to interpret). All of our transmission data for each sample are superimposable (Appendix A), which means that there is no deattenuation or depolarisation.

The output from the MMP measurements clearly indicated that there was true natural optical activity (i.e., CD) formed from the achiral starting monomer at all concentrations tested. These data indicate that there are significant circular dichroic contributions as well as linear effects that give rise to the overall CD spectrum observed with benchtop spectrometers. The magnitude of the CD signal acquired using the MMP method is large enough to allow us to conclude that the *J*-aggregates are helical in nature, thus chiral. The opposite-sign spectra observed from the solutions stirred CW and CCW imply that the stirring vortex and its directionality are responsible for the opposite chirality of the *J*-aggregates in these solutions. The relative intensities of the CD and LD signals acquired on a benchtop spectrometer are comparable with the corresponding ones acquired on the MMP.

Full MMP analysis of unstirred solutions at 15, 25 and 35 µM concentration displayed no CD and little-to-no LD. The absorption spectra for these solutions contained a significant amount of monomer as indicated by the strong 504 nm band (Appendix A).

The full results for the MMP work are summarised in Table 2. The MMP analysis confirms that CW and CCW stirring led to solutions containing *J*-aggregates of opposite chirality, which possess intrinsic chirality. The difference in the dissymmetry factor between stirred and unstirred solutions highlights that the chirality of the former is due to the vortex produced mechanically in solution.

#### Films of *J*-aggregates

One of the key uses of cyanine dyes is in the generation of thin films for use in plasmonics [20]. Typically, these films are doped with a chiral molecule which transfers chirality to the supramolecular structure during the formation of aggregates. For this study, we wanted to test whether the chiral *J*-aggregates formed while stirring solutions of S0271 are present in dropcasted thin films on fused silica-substrate plates. All the thin films were produced by dropcasting 40 μL of each stock solution on a clean fused silica plate, followed by air drying under constant airflow in a fume cupboard. The fused silica plates were centred in the MMP sample holder, and scans in CD and MMP modes of the central point of the sample were recorded at 24 °C. In the MMP mode, a 5 × 5 grid at 0.5 mm steps was scanned at a single wavelength of the thin film, taking advantage of the highly collimated beam at B23. These measurements were used to assess the homogeneity of the film in terms of optical and chiroptical properties of the amount of material deposited. Two films were analysed: they were produced from a CW-stirred solution (35 µM in Milli-Q water, pH 7.12, 18.2 Ω cm^−1^) and a CCW-stirred solution (15 µM in Milli-Q water, pH 7.12, 18.2 Ω cm^−1^).

In both films made from a CW- and CCW-stirred solution, respectively, the *J*-aggregate band was present in the spectrum (Figure 7a), albeit significantly reduced when compared to the corresponding *J*-aggregate band in the solution (Table 2). The MMP data suggested that the chirality of the *J*-aggregates present in the solution has been lost when the solution was transferred to the fused silica plate and air dried. While this is disappointing, it is not surprising as solvent evaporation has a large effect on supramolecular polymers with relatively small association constants between the monomers (see Section 2.2). The maximum CD intensity was −4.34 mdeg for the films obtained from the CW-stirred solution and 2.25 mdeg for those obtained from the CCW-stirred solutions. Both values are 4–10-fold lower than the corresponding data in solution.

In order to analyse this further, mapping of the film was performed at 562 nm (λ_max_
*J*-aggregate), which is shown in Figure 7. MMP mapping [31,32] of a 5 × 5 grid (Figure 7b) revealed that while the films were relatively uniform in terms of absorbance (Abs map), the CD and CB intensities were very small and close to the limit of detection of the instrument. On the other hand, the LD and LB are significantly larger, implying that the refractive index has significantly changed when shifting from solutions to films. Due to the anisotropy of the films generated from dropcasting, the most relevant measure of chirality is the g-factor, which was not greater than 10^−3^.

## 3. Materials and Methods

### 3.1. Solution and Film Generation

The S0271 dye was received from Few chemicals and used without further purification. The solutions were made by dissolving the dye in the relevant volume of ultrapure Milli-Q^®^ water (pH 7.12, 18.2 Ω cm^−1^) before sonication for 5 min to ensure full dissolution. Some 7.5 mL were taken from the central stock and placed into a 25 mL conical sample vial (dimensions [LxØ] 70.0 × 27.0 mm) along with an elongated cylindrical magnetic stirrer bar (dimensions [LxØ] 13.0 × 5.0 mm). These were immediately set stirring on an IKA^®^ RCT Basic (CW stirring) and a Heidolph^®^ MR Hei-Standard (CCW stirring) at 500 rpm, at 21 °C for the benchtop studies and 24 °C for the MMP studies.

Films were made via dropcasting 40 μL of a stirred solution onto a quartz plate and leaving to dry at 24 °C.

### 3.2. CD and MMP Studies

For the CD and MMP experiments, aliquots were taken from these samples without stopping the stirring of the solutions and tested in 5-mm-pathlength (benchtop CD) or 10-mm-pathlength (MMP) cylindrical cuvettes. In each instance, the solutions were diluted as required to reduce the absorbance below the maxima for the spectrometers. The MMP data were processed using the Matrix logarithm method [23].

#### 3.2.1. Benchtop CD Studies

CD and UV-Vis experiments were performed on an Chirascan^®^ Circular Dichroism Spectropolarimeter (Applied Photophysics, Leatherhead, UK) or a J-810 Spectropolarimeter (Jasco, Tokyo, Japan) equipped with a Peltier temperature controller. The following parameters were used for full-spectra measurements: wavelength-scanning range, 450–600 nm; temperature, 21 °C; scanning increments, 1 nm; monochromator bandwidth, 2 nm; sampling time per point, 1 s. The flow LD studies were completed using an SJ-1211H Atto^®^ Perista-Pump^®^ (ATTO corporation, Tokyo, Japan), using a 3 mm I.D. and 5 mm O.D. tubing with the flow rates defined on the analogue dial. All CD data were processed using OriginPro^®^ 2021b (OriginLab Corporation, Northampton, MA, USA).

#### 3.2.2. MMP Studies

CD, MMP and UV-Vis experiments were performed on the B23 beamline at the Diamond Light Source (Didcot, UK), during the awarded beamtimes SM29075 and SM31797, using a Mueller matrix polarimeter (MMP) tower on B23 module B [30]. The following parameters were used for full-spectra measurements for only CD and UV-Vis involving solutions: wavelength-scanning range, 450–600 nm; temperature, 24 °C; scanning increments, 1 nm; integration time, 0.05 s; 16 integration cycles; scan repeats, 1; 10 mm pathlength. For full deconvolution using the MMP with solutions, the following parameters were used: wavelength-scanning range, 450–600 nm; temperature, 24 °C; scanning increments, 1 nm; integration time, 0.05 s; 16 integration cycles; scan repeats, 4. The absorbance intensity had to be ≤1.6 to allow enough light to reach the detector. These showed the same directionality with respect to stirring direction with the solutions tested on the benchtop spectrometers. The intensity of the CD response was of a similar scale to those measured on the J-810 spectropolarimeter. For the MMP analysis, a CD signal of ≥50 mdeg was required for a good signal-to-noise ratio for full deconvolution. With the solutions tested on the MMP, the measured intensity was sufficiently high enough to perform MMP analysis.

The following parameters were used for the films: wavelength-scanning range, 450–600 nm; temperature, 24 °C; scanning increments, 1 nm; 4 × 4 or 5 × 5 matrix of points measured; integration time, 0.05 s; 16 integration cycles; scan repeats, 1.

All MMP and CD data were processed using MMP converter and OriginPro^®^ 2021b using scripts designed by T.M.G. for analysis of the individual matrices.

## 4. Conclusions

Within this study, we have shown that the optical activity of *J*-aggregates is dictated by the direction of the vortex: clockwise or counterclockwise. Induced mechanochirality of this type is unusual but fits within the scope of data reported in the literature [12,13,28,33]. We have also demonstrated that the CD observed with commercial spectropolarimeters has a component that is assigned to true natural optical activity and is not entirely LD from the formation of quasi-unidimensional large supramolecular structures.

The MMP has shown that there is a large LD contribution to the chiral response of *J*-aggregates. This is naturally expected for supramolecular aggregates that align with linear-polarised light which also showed a strong contribution of LB. However, despite the sizeable contribution of LD and LB, there is an unambiguous CD contribution in the solution of *J*-aggregates as decomposed from the differential MMP data. The synchrotron MMP data using the B23 beamline were crucial for this study as they elucidated the natural optical activity of the *J*-aggregates in solution. The shear forces caused by stirring the solution are at the origin of the formation of chiral *J*-aggregates in S0271. These forces drive the monomers to stack in a twisted arrangement leading to a helical aggregate, which is the origin of the chiroptical activity measured with B23 MMP.

Solvent is important for the stability of the *J*-aggregates, with the hydrophobic effect exerted on the π-surfaces of S0271 being the driver for self-assembly of the monomers. The loss of solvent during the formation of thin films leads to significant destruction of the chiral *J*-aggregates.

This study paves the way for other researchers working with quasi-unidimensional molecules/aggregates to investigate acquired optical activity by stirring using the MMP facility at the Diamond B23 beamline to better understand the relationship between CD and LD for these types of systems. It also makes significant progress towards the use of induced chiral *J*-aggregates as alternatives to plasmonics as it provides unprecedented insights into the polymerisation mechanism, also clarifying the origins of the optical activity of such supramolecular aggregates.

## Figures and Tables

**Figure 1 molecules-28-01523-f001:**
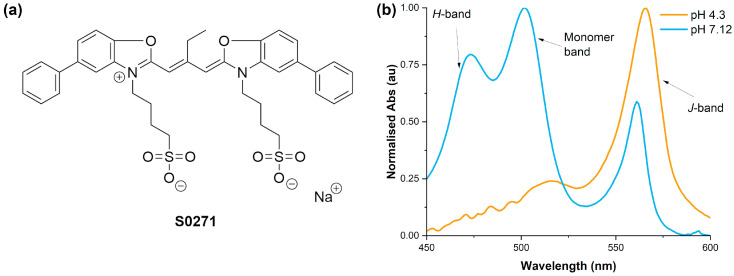
(**a**) The structure of the cyanine dye S0271 used in this study. (**b**) Two normalised visible spectra of S0271 solutions at different pHs which highlight the absorption bands of the monomer and *H*- and *J*-aggregates.

**Figure 2 molecules-28-01523-f002:**
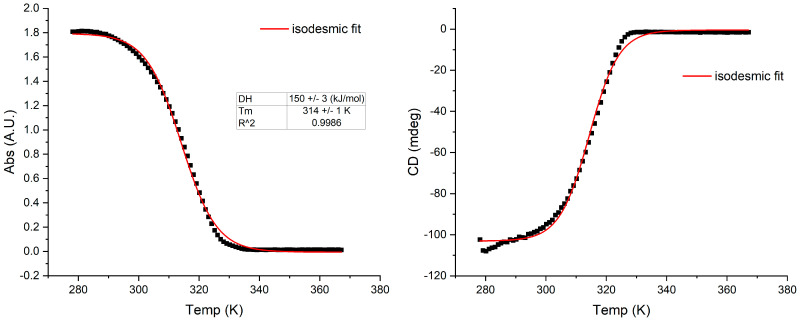
The melting curves of *J*-aggregates of S0271 at 20 µM in Milli-Q water; left: absorption, right: CD (pH 6.7, 18.2 Ω cm^−1^) from clockwise stirring, monitored at 562 nm, recorded at 21 °C. The thermodynamic functions were calculated with an isodesmic polymerisation model, see methods. ΔH = enthalpy of association; Tm = melting temperature at which 50% of the aggregate is dissociated.

**Figure 3 molecules-28-01523-f003:**
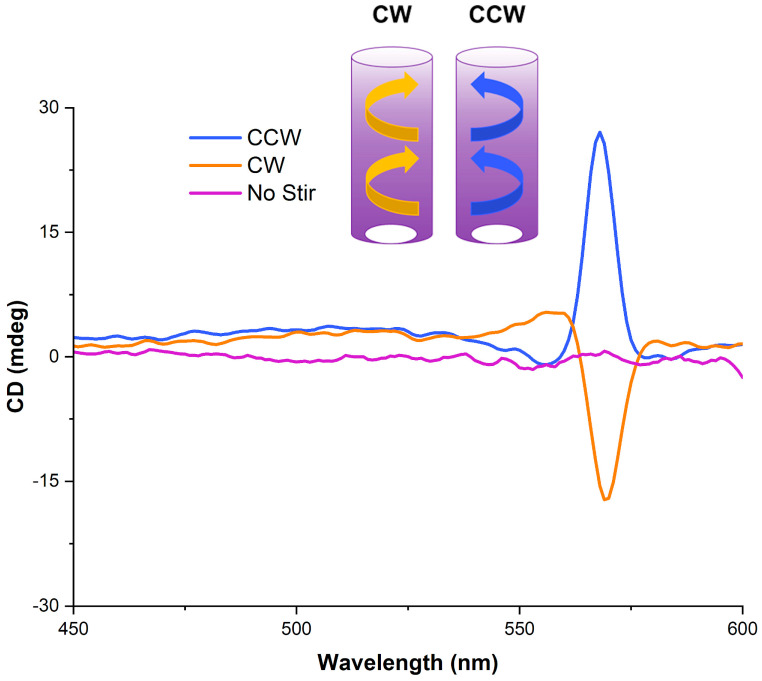
CD spectra, recorded at 21 °C, from stirring solutions of S0271 at 25 µM in Milli-Q water (pH 7.12, 18.2 Ω cm^−1^) in opposite directions (CW—orange and CCW—blue), whilst the non-stirred trace is shown in purple.

**Figure 4 molecules-28-01523-f004:**
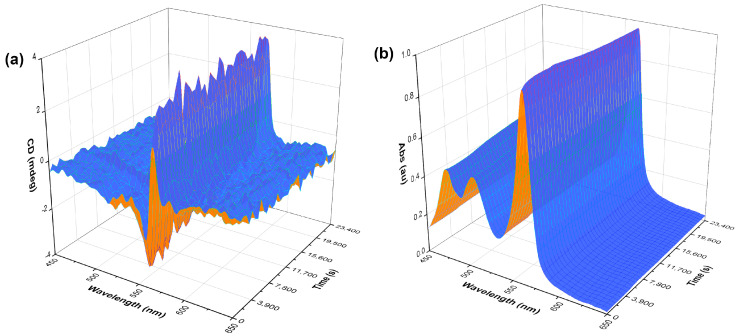
A three-dimensional kinetics plot of S0271 at 40 µM in Milli-Q water (pH 7.12, 18.2 Ω cm^−1^) stirred CW in situ in the CD spectrometer. (**a**) CD kinetics spectra and (**b**) UV-vis kinetics spectra. The measurements were taken across 6.5 h taking a measurement every 10 min, recorded at 21 °C.

**Figure 5 molecules-28-01523-f005:**
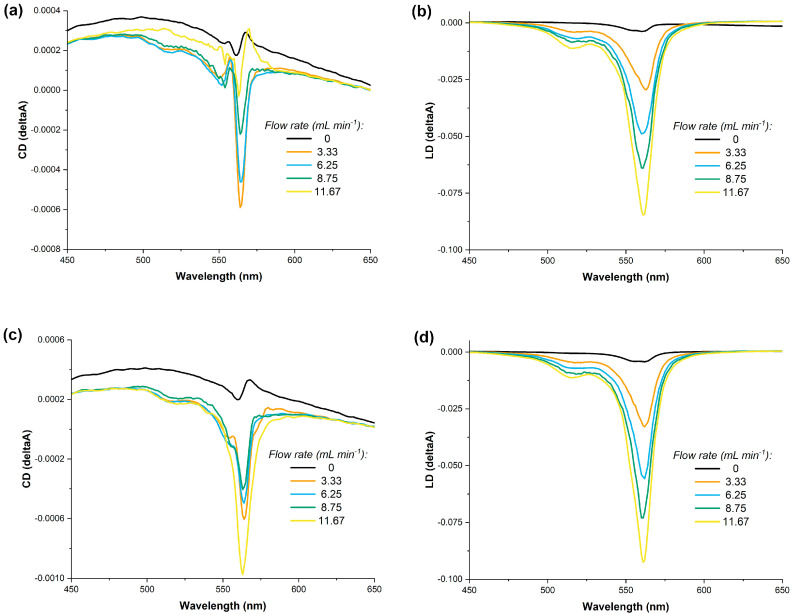
The dependence of the LD and CD of a solution of S0271 in Milli-Q water (pH 7.12, 18.2 Ω cm^−1^) at 20 µM on the rate of a constant laminar flow, panels (**a**,**b**) no stirring and, panels (**c**,**d**) with stirring in situ at 400 rpm, all recorded at 21 °C.

**Figure 6 molecules-28-01523-f006:**
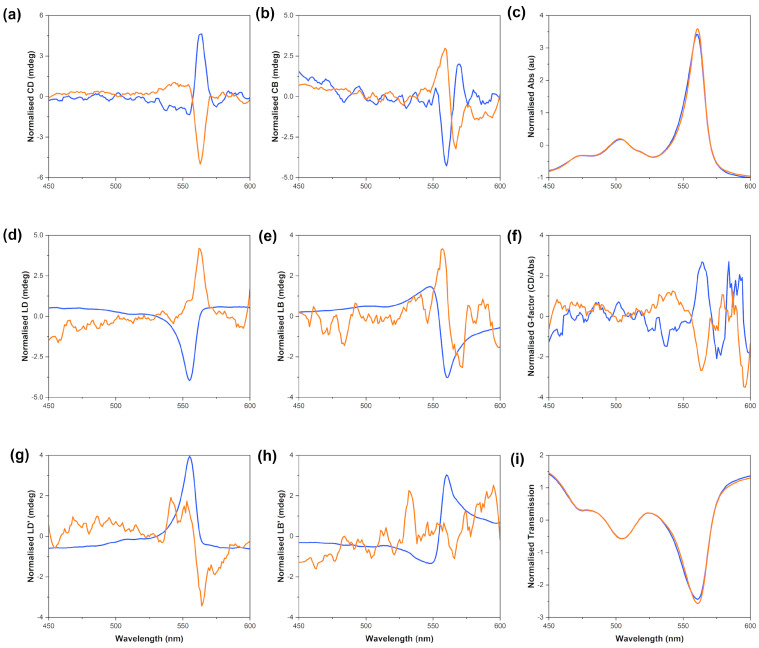
S0271 *J*-aggregates scanned on the MMP; *J*-aggregates assembled by stirring 25 µM solutions in Milli-Q water (pH 7.12, 18.2 Ω cm^−1^) CW (orange) and CCW (blue) for 16 h before aliquoting, recorded at 24 °C. The natural optical activity is highlighted in graph (**a**), (**b**) circular birefringence, (**c**) absorbance, (**d**) linear dichroism, (**e**) linear birefringence, (**f**) g_abs_, (**g**) 45° linear dichroism, (**h**) 45° linear birefringence, (**i**) transmission. The data shown in this figure are the average of four scans, smoothed and normalised using Z-scores.

**Figure 7 molecules-28-01523-f007:**
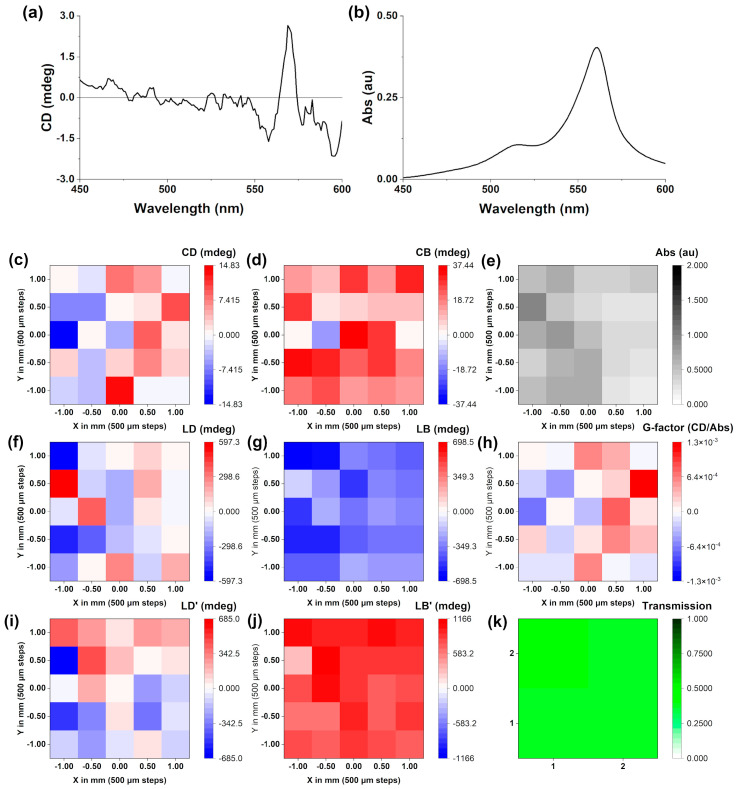
(**a**) The CD scan of a *J*-aggregate dropcasted film made from a 35 µM CW-stirred solution in Milli-Q water (pH 7.12, 18.2 Ω cm^−1^). A matrix map plot generated from a 5 × 5 matrix of points picked at 562 nm, generated from the same film showing: (**a**) circular dichroism, (**b**) UV-Vis absorption, (**c**) circular dichroism (**d**) circular birefringence, (**e**) UV-Vis absorption, (**f**) linear dichroism, (**g**) linear birefringence, (**h**) g_abs_, (**i**) 45° linear dichroism, (**j**) 45° linear birefringence, (**k**) transmission. All recorded at 24 °C.

**Table 1 molecules-28-01523-t001:** The dissymmetry-factor dependence as function of the solution concentration measured on a benchtop CD spectrometer at 15 µM, 25 µM, 35 µM, 50 µM and 75 µM in Milli-Q water (pH 7.12, 18.2 Ω cm^−1^) at the wavelength of maximum ellipticity, θ_max_, ca. 564 nm, recorded at 21 °C.

Solution Conc. (µM)	CW g_abs_ (×10^−3^)	CCW g_abs_ (×10^−3^) ^1^
15	−2.52	2.13
25	−2.11	2.58
35	−2.43	3.27
50	−2.42	2.00
75	−1.19	1.69

^1^ CW stirring rate 500 rpm (digital control), CCW stirring rate 500 ± 50 rpm (analogue control).

**Table 2 molecules-28-01523-t002:** Solution studies: the CD and dissymmetry factor dependence as function of stirring direction and solution concentration measured on MMP at 15 µM, 25 µM and 35 µM in Milli-Q water (pH 7.12, 18.2 Ω cm^−1^) at the wavelength of maximum ellipticity, θ_max_, average of 4 scans, recorded at 24 °C.

Solution Conc. (µM)	CW		CCW		No Stirring	
	CD (mdeg)	g_abs_ (×10^−3^)	CD (mdeg)	g_abs_ (×10^−3^)	CD (mdeg)	g_abs_ (×10^−3^)
15	−27.77	−1.46	45.37	1.00	1.21	−0.027
25	−30.09	−0.91	22.56	0.68	0.24	0.023
35	−16.47	−1.03	67.50 ^1^	2.41 ^1^	13.28	0.33

^1^ A single scan was run for this sample.

## Data Availability

All data underlying the findings of this work are available from the corresponding author upon reasonable request.

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
