# Peer review of "Mueller Matrix Polarimetry on Cyanine Dye J-Aggregates"

_molecules, 2023, doi:10.3390/molecules28041523_

Round 1
Reviewer 1 Report
In this manuscript, through vortex induction, the authors put forward a method and mechanism for assembling J-aggregates of cyanine dyes, and pointed out that the direction of vortex (such as CW and CCW) was the key reason affecting the chirality of J-aggregates. In addition to the traditional desktop CD spectrometer, the author also used Mueller matrix polarimetry to study the chiral characteristics of J-aggregates, including CD, CB, LB, LD, LD', LB', gabs and so on, showing the properties of cyanine dye J-aggregates more accurately. However, there are still some issues in the manuscript that need to be revised carefully. A major revision is needed to make the manuscript more perfect.
1. The experimental condition in Figure 4a is CW vortex, and its CD spectrum shows two obvious signal peaks, while in Figure 3, only a single signal peak appears under the same conditions. The results of the two are obviously different, please explain. If necessary, the explanation can be displayed in the manuscript.
2. Page 5, Line 138-151. According to previous research, it is found that cyanine dyes are not stable enough, which are easy to be discolored and invalidated after a long time. The authors said, “The solutions retained the chiral J-aggregates even after 5 days, regardless whether they were left stirring or not.” Please give evidence and preferably list it in Supplementary Materials.
3. The unit of “Max” in the caption of Figure 5 is incorrect. Please check.
4. Do not descript the experimental temperature in general terms r.t. Please provide the specific value.
5. In the manuscript, it is better to point out the corresponding position of the figures of Supplementary Materials, otherwise so many supplementary figures will cause confusion to the reader.
6. “Conclusion” section is necessary, please add.
7. Please carefully check all references and some old literatures might be updated, especially those in Introduction.
Author Response
Reviewer 1
In this manuscript, through vortex induction, the authors put forward a method and mechanism for assembling J-aggregates of cyanine dyes, and pointed out that the direction of vortex (such as CW and CCW) was the key reason affecting the chirality of J-aggregates. In addition to the traditional desktop CD spectrometer, the author also used Mueller matrix polarimetry to study the chiral characteristics of J-aggregates, including CD, CB, LB, LD, LD', LB', gabs and so on, showing the properties of cyanine dye J-aggregates more accurately. However, there are still some issues in the manuscript that need to be revised carefully. A major revision is needed to make the manuscript more perfect.
We thank the reviewer for taking the time to read and assess our manuscript. We have made comments under each of the points raised by the reviewer and, where necessary, we have amended the manuscript accordingly.
- The experimental condition in Figure 4a is CW vortex, and its CD spectrum shows two obvious signal peaks, while in Figure 3, only a single signal peak appears under the same conditions. The results of the two are obviously different, please explain. If necessary, the explanation can be displayed in the manuscript.
We agree with the reviewer. This difference is due to the different experimental conditions between the standard set of experiments and the kinetics study (rectangular cuvette, 4mm pathlength, rice-sized stir-bar). We have added a comment at line 146 describing this.
- Page 5, Line 138-151. According to previous research, it is found that cyanine dyes are not stable enough, which are easy to be discolored and invalidated after a long time. The authors said, “The solutions retained the chiral J-aggregates even after 5 days, regardless whether they were left stirring or not.” Please give evidence and preferably list it in Supplementary Materials.
We thank the reviewer for this comment, we agree that the J-aggregates of some cyanine dyes are not stable. We have observed this in our lab with other cyanine dyes, however S0271 makes long lasting aggregates. We added figure S10 which shows the behaviour of the J-aggregate solutions od S0271 after 5 days, with and without stirring.
- The unit of “Max” in the caption of Figure 5 is incorrect. Please check.
We thank the reviewer for this comment, Figure 5 was amended to show the flow rates in mL min-1.
- Do not descript the experimental temperature in general terms r.t. Please provide the specific value.
We thank the reviewer for this comment, we added in the captions of all figures and tables the exact temperature for each experiment.
- In the manuscript, it is better to point out the corresponding position of the figures of Supplementary Materials, otherwise so many supplementary figures will cause confusion to the reader.
We thank the reviewer for this comment, we have amended the manuscript accordingly.
- “Conclusion” section is necessary, please add.
We thank the reviewer for pointing this out. We have incorrectly labelled the Conclusion section as Discussion in the original manuscript. We have now corrected this and hopefully this is now clearer.
- Please carefully check all references and some old literatures might be updated, especially those in Introduction.
We thank the reviewer for this comment, more recent and general literature examples have been added in the introduction.
Reviewer 2 Report
The manuscript by Pantos and coworkers reported the polymerization mechanism of a chiral J-aggregate formed by cyanine dye S0271. The authors discussed the optical properties of S0271 aggregates under different flow dynamics by benchtop spectropolarimeters and Mueller matrix polarimetry, unveiling the influence of shear forces on the formation of J-aggregates. The manuscript may be suitable to be published at Molecules after some further revision according to the detailed comments below:
Questions:
1) Introduction: Some previous research on J- and/or H-aggregates of cyanine dyes, for example, Nature 1936, 138, 1009; Nature Chem. 2012, 4, 655; CCS Chem. 2021, 3, 678; J. Am. Chem. Soc. 2014, 136, 28; J. Phys. Chem. Lett. 2022, 13, 508; Aggregate 2022, e261, should be included.
2) The author assigned the red-shifted absorption band at ~ 560 nm as J-band. Is there any proof for the assignment? If not, additional proof is needed to confirm the J-band such as steady-state fluorescence or fluorescence lifetime. In addition, the absorption spectrum of S0271 monomers should be included in Figure 1.
3) Figure 1b showed the influence of pH (pH = 4.3 and 6.7) on the formation of J-aggregates. However, why were the rest tests carried out at pH = 7.12?
4) The CD spectrums of S0271 stirred by CW and CCW showed the chirality of J-aggregates was opposite. Is there any solid proof, such as 2D NMR or XRD results of the chirality?
Author Response
The manuscript by Pantos and coworkers reported the polymerization mechanism of a chiral J-aggregate formed by cyanine dye S0271. The authors discussed the optical properties of S0271 aggregates under different flow dynamics by benchtop spectropolarimeters and Mueller matrix polarimetry, unveiling the influence of shear forces on the formation of J-aggregates. The manuscript may be suitable to be published at Molecules after some further revision according to the detailed comments below:
We thank the reviewer for taking the time to read and assess our manuscript. We have made comments under each of the points raised by the reviewer and, where necessary, we have amended the manuscript accordingly.
Questions:
- Introduction: Some previous research on J- and/or H-aggregates of cyanine dyes, for example, Nature 1936, 138, 1009; Nature Chem. 2012, 4, 655; CCS Chem. 2021, 3, 678; J. Am. Chem. Soc. 2014, 136, 28; J. Phys. Chem. Lett. 2022, 13, 508; Aggregate 2022, e261, should be included.
We thank the reviewer for highlighting these articles, they are all relevant and have been added as references.
- The author assigned the red-shifted absorption band at ~ 560 nm as J-band. Is there any proof for the assignment? If not, additional proof is needed to confirm the J-band such as steady-state fluorescence or fluorescence lifetime. In addition, the absorption spectrum of S0271 monomers should be included in Figure 1.
We thank the reviewer for this comment, however we have made the J-aggregate band assignment based on literature precedents, references 2, 3, 4 and 6 in the revised manuscript. The absorption spectrum of the pure monomer would need to be done in a different solvent (MeOH or EtOH) which will not be a good comparison to the rest of the study. We have used a milli-Q water solution (pH 7.12) to show the main species, H-aggregate, monomer and J-aggregate; we believe this is a better reference point for our study than a solution of S0271 in a different solvent.
- Figure 1b showed the influence of pH (pH = 4.3 and 6.7) on the formation of J-aggregates. However, why were the rest tests carried out at pH = 7.12?
We thank the reviewer for pointing this out. The source of the discrepancy was the source of milli-Q water. We have replaced the pH 6.7 data with a newer one recorded at pH 7.12 showing H-, monomer and J-aggregate bands.
- The CD spectrums of S0271 stirred by CW and CCW showed the chirality of J-aggregates was opposite. Is there any solid proof, such as 2D NMR or XRD results of the chirality?
We thank the reviewer for this comment however we believe that CD is the best way to prove that the aggregates have opposite chirality in solution. The symmetry of S0271 makes the 2D NMR data (nOe) ambiguous as it is impossible to distinguish between intra- and inter-molecular through space contacts. The XRD, while a very interesting and attractive suggestion is not going to be a good indication of the structures of the J-aggregates in solution as we know that desolvation leads to structural changes in the aggregates, as we showed in our thin-film studies, section 2.3.1 in the manuscript.